# 3DLAND: 3D Lesion Abdominal Anomaly Localization Dataset

## Abstract

Existing medical imaging datasets for abdominal CT often lack three-dimensional annotations, multi-organ coverage, or precise lesion-to-organ associations, hindering robust representation learning and clinical applications. To address this gap, we introduce 3DLAND, a large-scale benchmark dataset comprising over 6,000 contrast-enhanced CT volumes with over 20,000 high-fidelity 3D lesion annotations linked to seven abdominal organs: liver, kidneys, pancreas, spleen, stomach, and gallbladder. Our streamlined three-phase pipeline integrates automated spatial reasoning, prompt-optimized 2D segmentation, and memory-guided 3D propagation, validated by expert radiologists with surface dice scores exceeding 0.75. By providing diverse lesion types and patient demographics, 3DLAND enables scalable evaluation of anomaly detection, localization, and cross-organ transfer learning for medical AI. Our dataset establishes a new benchmark for evaluating organ-aware 3D segmentation models, paving the way for advancements in healthcare-oriented AI.

## 1 Introduction

Abdominal computed tomography (CT) is a cornerstone of clinical diagnostics, enabling precise identification and monitoring of lesions across multiple organs to support treatment planning and longitudinal care. However, progress in *anomaly detection* (identifying lesions) and *anomaly localization* (pinpointing their precise location) is constrained by the scarcity of large-scale, multi-organ, three-dimensional datasets with reliable lesion annotations. Escalating clinical demand, coupled with limited radiology resources, amplifies the need for consistently annotated data to drive robust representation learning (Cardoso et al., 2022).

Existing datasets have advanced medical imaging yet fall short in critical aspects. Single-organ benchmarks, such as those focused on liver or kidney (Bilic et al., 2023; Heller et al., 2020; Antonelli et al., 2022), limit generalization across abdominal anatomy and risk overlooking cross-organ pathologies. Multi-organ efforts broaden anatomical scope (Wasserthal et al., 2022; Ma et al., 2023; Zhao et al., 2023a) but typically lack pixel-level lesion annotations or explicit lesion-to-organ associations. Other resources provide 2D annotations without volumetric context or organ-aware linkage, hindering comprehensive clinical reasoning and robust evaluation of 3D segmentation models.

Three core challenges explain the absence of such a resource. *First*, lesions exhibit significant variability in appearance and context across organs (e.g., liver, kidneys, pancreas, spleen, stomach, gallbladder), rendering single-organ or coarse annotations inadequate for capturing distribution shifts. *Second*, the mismatch between 2D bounding boxes and voxel-accurate 3D masks under-specifies boundaries, limiting representation learning and faithful volumetric evaluation. *Third*, clinical ambiguities, such as inflammatory interfaces or imaging artifacts, necessitate streamlined, auditable pipelines with expert validation to ensure reliability.

To address these challenges, we introduce 3DLAND, the first large-scale benchmark dataset and pipeline for organ-aware 3D lesion segmentation in contrast-enhanced abdominal CT. Spanning over 20,000 2D lesions, 6,000 CT volumes and covering seven organs, 3DLAND provides high-fidelity volumetric lesion masks for diverse lesion types (e.g., cysts, tumors), validated by expert radiologists with Surface dice scores exceeding 0.799 in 2D and 0.752 in 3D lesion segmentation. As illustrated in Fig. 1, three important properties of our dataset are shown.

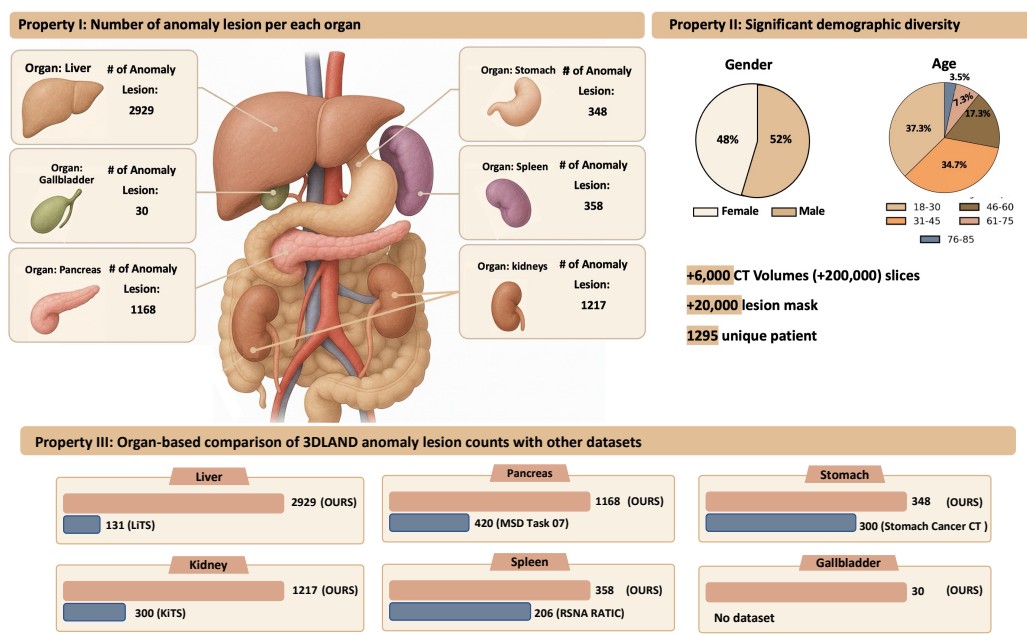

Figure 1: Comprehensive overview of 3DLAND dataset properties, highlighting its superiority in anomaly lesion coverage and demographic diversity.**Property I**: Number of anomaly lesions per abdominal organ, visualized with a 3D anatomical model showing lesion counts for liver, gallbladder, pancreas, spleen, kidneys, and stomach.**Property II**: Significant demographic diversity, including gender distribution and age groups, alongside key statistics.**Property III:** Organ-based comparison of 3DLAND anomaly 3D lesion counts with the largest datasets per each organ, demonstrating 3DLAND's extensive coverage across organs. This figure underscores 3DLAND's value as a large-scale, organ-aware benchmark for abdominal CT anomaly localization and downstream ML tasks.

Our contributions are:

- The first and largest dataset with organ-aware 3D lesion annotations for abdomen CT scans, enabling scalable benchmarking for anomaly detection and localization.
- A streamlined three-phase pipeline integrating spatial reasoning, prompt-optimized 2D segmentation, and memory-guided 3D propagation, with expert adjudication for clinical reliability.
- A benchmark of the state-of-the-art lesion segmentation models.
- A publicly available resource fostering advancements in representation learning, transfer learning, and cross-organ analysis, with code and data released under an open license.

In the following sections, we detail the dataset construction, validation protocols, and benchmarks, alongside comprehensive comparisons with prior datasets and methods.

## 2 RELATED WORK

Research in abdominal lesion segmentation has progressed along two primary axes: annotated datasets, which provide the foundation for model development and evaluation, and segmentation models, which leverage these datasets for automated lesion localization.

### 2.1 DATASETS

Early efforts in abdominal lesion segmentation focused on single-organ datasets. For instance, **LiTS** (Bilic et al., 2023) provides extensive liver tumor annotations, while **KiTS** (Heller et al., 2020) offers

Table 1: Comparison of abdominal CT datasets with lesion segmentation and expert validation.

| Dataset | # of Organs | # of CT | 2D Seg | 3D Seg | Expert Val |
|---|---|---|---|---|---|
| **Pancreas-CT** Roth et al. (2015) | 1 | 82 | ✗ | ✗ | ✓ |
| **LiTS** Bilic et al. (2023) | 2 | 201 | ✓ | ✓ | ✓ |
| **KiTS** Heller et al. (2020) | 2 | 300 | ✓ | ✓ | ✓ |
| **AbdomenCT-1K** Ma et al. (2023) | 4 | 1,000 | ✗ | ✗ | ✗ |
| **CT-ORG** Fang et al. (2020) | 4 | 140 | ✗ | ✗ | ✗ |
| **CHAOS** Kavur et al. (2021) | 4 | 40 | ✗ | ✗ | ✗ |
| **MSD CT Tasks** Antonelli et al. (2022) | 9 | 947 | ✓ | ✓ | ✓ |
| **RSNA RATIC** Rudie et al. (2024) | 5 | 4,274 | ✓ | ✗ | ✓ |
| **BTCV** Landman et al. (2015) | 13 | 50 | ✗ | ✗ | ✓ |
| **AMOS22** Ji et al. (2022) | 15 | 500 | ✗ | ✗ | ✓ |
| **Abdomen Atlas-8k** Zhao et al. (2023a) | 26 | 8,022 | ✗ | ✗ | ✓ |
| **WORD** Zhao et al. (2023b) | 16 | 150 | ✗ | ✗ | ✗ |
| **MSWAL** Wang et al. (2024) | 9 | 300 | ✓ | ✓ | ✓ |
| **ULS23** de Grauw et al. (2025) | 12 | 3,000 | ✓ | ✓ | ✗ |
| **DeepLesion** Yan et al. (2018) | – | 32,000 | ✗ | ✗ | ✗ |
| **3DLAND (Ours)** | **7** | **6,000** | ✓ | ✓ | ✓ |

labeled kidney scans. These datasets have significantly advanced organ-specific segmentation but are limited in generalizing across diverse abdominal anatomy or capturing cross-organ pathologies. Similarly, the **MSD** collection (Antonelli et al., 2022) supports multi-organ segmentation tasks but lacks lesion-level annotations, restricting its utility for anomaly detection and localization.

Recent multi-organ datasets, such as **TotalSegmentator** (Wasserthal et al., 2022), **AbdomenCT-1K** (Ma et al., 2023), and **AbdomenAtlas-8K** (Zhao et al., 2023a), broaden anatomical coverage by segmenting multiple organs. However, these datasets typically do not include pixel-level lesion annotations or explicit lesion-to-organ mappings, which are critical for clinical tasks like precise anomaly localization. **DeepLesion** (Yan et al., 2018), the largest public lesion dataset, comprises over 32,000 2D bounding box annotations across 10,000+ CT studies. Despite its scale, DeepLesion lacks 3D masks and organ-aware annotations, limiting its suitability for volumetric analysis or organ-specific tasks (e.g., distinguishing left vs. right kidneys).

More recent efforts, such as the **ULS23** challenge (de Grauw et al., 2025), advance 3D lesion segmentation by providing a universal benchmark. However, ULS23 does not offer explicit lesion-to-organ associations, hindering cross-organ analysis and clinical reasoning. Table 1 summarizes these datasets, highlighting 3DLAND's unique combination of multi-organ coverage, 3D lesion annotations, and organ-aware linkage across diverse lesion types (e.g., cysts, tumors) and patient demographics.

## 2.2 SEGMENTATION MODELS

Segmentation models for abdominal lesions have evolved from weak supervision to sophisticated 3D approaches. Early methods, such as **LesaNet** (Yan et al., 2019) and **MULAN** (Zhou et al., 2021b), relied on 2D bounding boxes or multi-view learning, achieving limited voxel-level precision. More recent approaches, like **MVP-Net** (Zhou et al., 2021c) and **DA-SHT** (Wang et al., 2021), improve detection through multi-view feature integration and hierarchical training but lack explicit organ-aware grounding, reducing their clinical applicability.

Prompt-based models, such as **MedSAM2** (Ma et al., 2025), enable 3D segmentation without extensive manual annotations, offering significant improvements in accuracy. However, these models do not inherently link lesions to specific organs, limiting their utility for tasks requiring anatomical context, such as cross-organ disease analysis. In contrast, our pipeline integrates organ segmentation with prompt-guided 3D lesion reconstruction, leveraging spatial reasoning and expert validation to produce clinically relevant, organ-aware annotations. This approach supports robust representation learning and transfer learning for downstream tasks like anomaly detection and lesion retrieval.

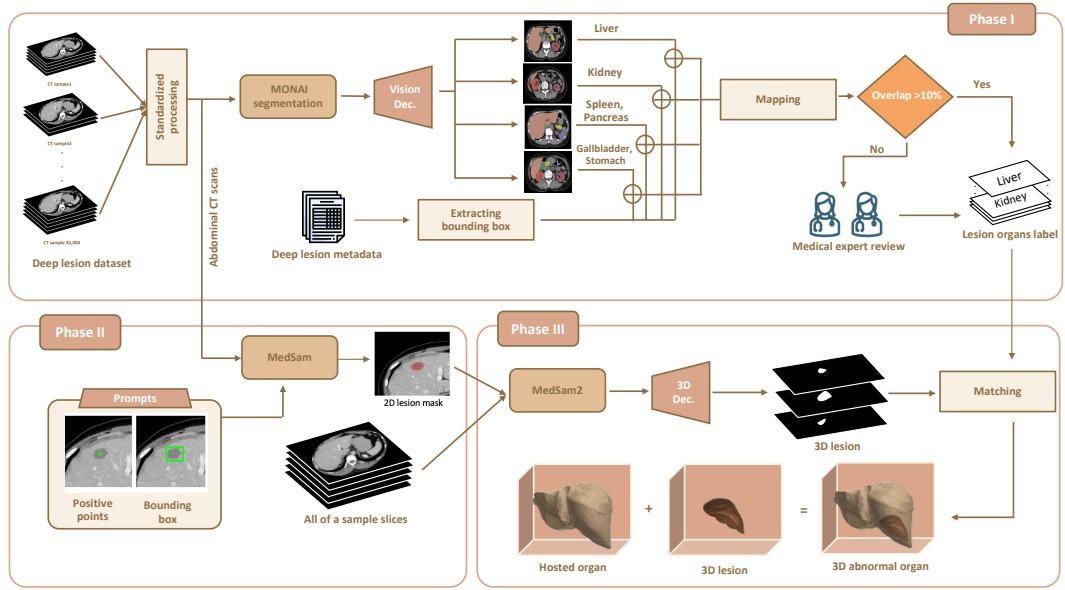

Figure 2: Overview of our three-stage pipeline for generating organ-aware 3D lesion masks from DeepLesion CT data. Phase I assigns each lesion to its most likely abdominal organ using MONAI-based Cardoso et al. (2022) organ segmentation and spatial overlap reasoning. Phase II performs precise 2D segmentation using a prompt-optimized MedSAM1 Ma et al. (2024) model. Phase III leverages MedSAM2 Ma et al. (2025) to propagate key-slice masks across volume slices, resulting in anatomically consistent 3D lesion reconstructions.

## 3 METHODOLOGY

To construct 3DLAND, we developed a streamlined, three-phase pipeline for organ-aware 3D lesion segmentation in contrast-enhanced abdominal CT scans. Spanning over 6,000 volumes from the DeepLesion dataset (Yan et al., 2018), our pipeline integrates automated spatial reasoning, prompt-optimized segmentation, and memory-guided volumetric propagation, with rigorous expert validation to ensure clinical reliability. The dataset covers seven abdominal organs (liver, kidneys, pancreas, spleen, stomach, gallbladder) and diverse lesion types (e.g., cysts, tumors) across varied patient demographics (age 18–85, balanced gender distribution). Fig. 2 illustrates the pipeline, with each phase detailed below.

### 3.1 PHASE I: LESION-TO-ORGAN ASSIGNMENT

This phase assigns lesions to their host organs using spatial reasoning and expert adjudication, addressing the lack of organ-aware annotations in DeepLesion (Yan et al., 2018).

#### 3.1.1 DATA PREPROCESSING

We preprocess 6,000+ contrast-enhanced CT volumes (512×512 pixels, 0.5–1 mm slice thickness, 1–3 mm inter-slice spacing) using standardized Hounsfield unit normalization (window: [-150, 250] HU). Organs are segmented using pretrained MONAI models (Cardoso et al., 2022) fine-tuned on TotalSegmentator (Wasserthal et al., 2022) (batch size 8, Adam optimizer, 100 epochs, Dice loss). This yields binary masks for seven organs, achieving an average Dice score of 0.92 across organs, as validated on a 10% subset by two medical experts.

### 3.1.2 Spatial Reasoning and Assignment

Lesions are matched to organs via a two-step spatial strategy. First, we compute the Intersection-over-Union (IoU) between each lesion's 2D bounding box (from DeepLesion) and organ masks:

$$\text{IoU}(B, M) = \frac{\text{Area}(B \cap M)}{\text{Area}(B \cup M)}, \tag{1}$$

where $B$ is the bounding box and $M$ is the organ mask. Lesions with IoU $> 10\%$ are assigned to the corresponding organ. The 10% threshold was determined via ablation studies (see Figure 4a), balancing sensitivity and specificity, consistent with prior work (Yan et al., 2018). For ambiguous cases (IoU $\leq 10\%$), we compute the Euclidean distance from the lesion center $c$ to the nearest organ boundary:

$$d(c, M) = \min_{p \in M} \|c - p\|_2, \tag{2}$$

assigning the lesion to the closest organ within 20 pixels (approximately 10 mm, based on average CT resolution). This threshold was validated through experiments, which showed 95% assignment accuracy on a 10% subset. A medical expert manually reviewed all proximity-based assignments and ambiguous cases (10% of lesions). This 20 pixel threshold also was determined via ablation studies (see Figure 4b)

### 3.2 Phase II: 2D Lesion Segmentation

This phase generates precise 2D lesion masks from DeepLesion bounding boxes using prompt-optimized segmentation, mitigating over-segmentation issues.

We leverage the pre-trained **MedSAM1** model (Ma et al., 2024) for 2D lesion segmentation, using DeepLesion bounding boxes as initial prompts. To mitigate over-segmentation, we shrink boxes to 70% of their original size, a value optimized through ablation studies (see Fig. 3) to balance lesion coverage and boundary precision. A center point is incorporated as a single-pixel positive prompt to enhance semantic accuracy. This yields 2D masks with an average Dice score of 0.807, validated on a 10% subset by two medical experts. More information about these shrinking boxes and the ablation study is explained in 4.2.1

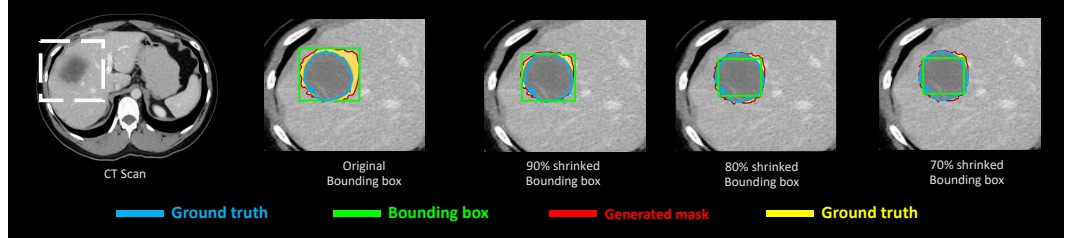

Figure 3: Effect of bounding box shrinkage on 2D lesion segmentation with MedSAM1. As the box size decreases from 100% to 70%, the pseudo mask (red) aligns more closely with the ground truth (blue), and the error region (yellow) is reduced. The 70% box provides the best trade-off between focus and coverage.

### 3.3 Phase III: 3D Lesion Segmentation and Evaluation

To generate voxel-accurate 3D lesion masks linked to their host organs, we leverage the pre-trained **MedSAM2** model (Ma et al., 2025) for inference, using 2D masks from Phase II as initial prompts. The DeepLesion dataset (Yan et al., 2018) provides slice ranges to define the volumetric extent. Each 2D mask is encoded into memory features capturing lesion appearance and spatial context, which MedSAM2 propagates across adjacent slices using its spatiotemporal attention mechanism. Inference is performed with a batch size of 8 on a NVIDIA A100 GPU, processing 6,000+ CT volumes in approximately 10 hours (0.05 GPU hours per volume), ensuring scalability for large-scale applications. This yields 3D masks with an average Surface dice score of 0.755, validated on a 10% subset (600 volumes) against manual annotations by an expert.

# 4 RESULTS

We present the results of constructing below, we detail validation metrics, ablation studies, error analysis, and baseline experiments, supported by comparisons to prior datasets and visualizations of annotation quality.

## 4.1 PHASE I: LESION-TO-ORGAN ASSIGNMENT

This phase leverages spatial reasoning to assign lesions to their host organs, enabling precise localization for clinical applications and downstream machine learning tasks like anomaly detection. Validation was performed through spatial metrics and blinded clinical review, with results detailed below.

### 4.1.1 ERROR ANALYSIS AND VISUALIZATION

To assess organ-aware lesion labeling, we conducted a blinded clinical review on 20% of lesions per organ, spanning both large (e.g., liver) and complex (e.g., gallbladder) cases. *Assignment Accuracy* measures the percentage of lesions correctly assigned to one of seven abdominal organs (liver, kidneys, pancreas, spleen, stomach, gallbladder) using spatial reasoning with IoU $> 10\%$ and Euclidean distance $< 20$ pixels. This metric evaluates the pipeline's precision in localizing lesions to their corresponding organs, which is critical for clinical applications (e.g., accurate lesion localization for diagnosis) and ML tasks (e.g., anomaly detection, representation learning). Higher accuracy reflects fewer misassignments, ensuring reliable organ-aware annotations for downstream tasks.

In this study, we prioritized models capable of segmenting all abdominal organs without requiring prompts to ensure an automated, end-to-end pipeline. We selected **MONAI** (Cardoso et al., 2022), a U-Net-based framework optimized for medical imaging, for its robust multi-organ segmentation capabilities and compatibility with our dataset (6,000+ volumes, 20,000+ 2D lesions). To benchmark our approach, we compared MONAI against other state-of-the-art (SOTA) models, including U-Net-based **nnU-Net** (Isensee et al., 2021), transformer-based **Swin UNETR** (Hatamizadeh et al., 2021), zero-shot **SAM 2** (Ma et al., 2024), and prompt-based **MedSAM** (Ma et al., 2024) and **MedSAM2** (Zhu et al., 2024). As shown in Table 2, MONAI was chosen for its competitive Dice scores (82.0–94.5%) with low variance (SD 1.2–1.8), demonstrating robustness across all organs. It outperforms competitors on larger organs (e.g., liver: $94.5 \pm 1.2$, kidneys: $94.0 \pm 1.3$) and achieves strong results on smaller organs (e.g., pancreas: $82.0 \pm 1.8$, gallbladder: $91.0 \pm 1.7$), with minimal user input. In contrast, models like **MedSAM** (Ma et al., 2024) lagged behind by 3–5%, particularly for complex cases (e.g., gallbladder: $85.0 \pm 2.8$). These results, combined with our high assignment accuracy (94.8% on a 5% subset), confirm the robustness of our spatial reasoning approach for organ-aware assignments, leveraging MONAI's U-Net-based architecture for high precision and automation.

Table 2: Comparison of accuracy of lesion to organ mapping based on multi-organ segmentation in abdominal CT scans across SOTA models. Our pipeline uses MONAI as the base model, achieving competitive performance with low variance across organs. Bold indicates the best score per organ; underline the second-best.

| Organ | nnU-Net | SwinUNETR | SAM 2 | MedSAM | MedSAM2 | MONAI |
|---|---|---|---|---|---|---|
| Liver | **95.0 ± 1.0** | 94.0 ± 1.5 | 72.0 ± 2.5 | 90.0 ± 2.0 | 92.0 ± 1.8 | 94.5 ± 1.2 |
| Kidneys | 92.0 ± 1.2 | 93.0 ± 1.7 | 71.4 ± 2.7 | 87.0 ± 2.2 | 90.0 ± 1.9 | **94.0 ± 1.3** |
| Pancreas | 80.0 ± 2.0 | 81.0 ± 2.2 | 62.0 ± 3.0 | 72.0 ± 3.5 | 75.0 ± 2.5 | **82.0 ± 1.8** |
| Spleen | **92.0 ± 1.1** | 91.0 ± 1.6 | 58.0 ± 2.8 | 85.0 ± 2.3 | 88.0 ± 2.0 | 91.5 ± 1.4 |
| Stomach | 88.0 ± 1.5 | 89.0 ± 1.8 | 61.6 ± 2.9 | 82.0 ± 2.4 | 85.0 ± 2.1 | **89.5 ± 1.5** |
| Gallbladder | 92.0 ± 2.0 | **93.0 ± 1.9** | 58.0 ± 3.2 | 85.0 ± 2.8 | 88.0 ± 2.3 | 91.0 ± 1.7 |

### 4.1.2 VALIDATION METRICS AND ABLATION STUDY

In **Phase I** (lesion-to-organ assignment), spatial reasoning using IoU ($> 10\%$) and Euclidean distance ($< 20$ pixels) achieved 94.8% assignment accuracy on a 10% subset (300 volumes), as validated by a medical expert. Figures 4a and 4b present ablation studies justifying these thresholds, showing that

IoU > 10% and distance < 20 pixels minimize ambiguous assignments (15%) while maintaining high accuracy.

We selected Assignment Accuracy to measure the percentage of lesions correctly assigned to one of seven abdominal organs, reflecting the pipeline's precision in organ-aware localization critical for clinical diagnosis and ML tasks like anomaly detection. Higher accuracy is preferred for reliable assignments. Ambiguous Cases quantifies lesions with unclear organ assignments due to overlap or proximity, impacting diagnostic specificity. Lower ambiguous cases are desirable to ensure clarity in organ identification. These metrics align with clinical needs and ML evaluation standards, balancing precision and specificity.

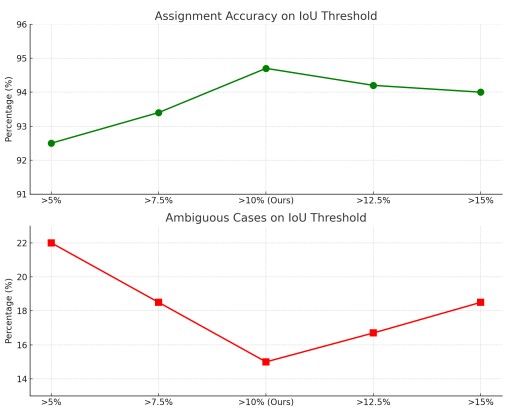 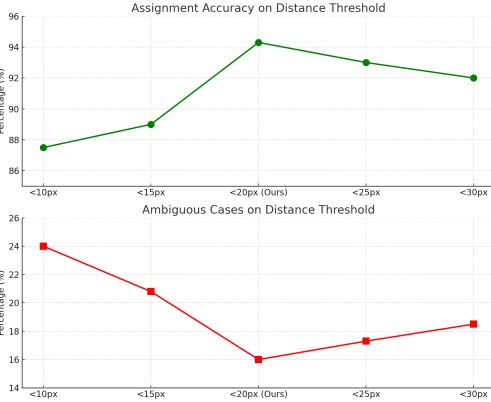

(a) **Figure 5.a** Ablation study on IoU thresholds, showing that IoU > 10% achieves the highest assignment accuracy (94.8%) with minimal ambiguous cases (15.4%).

(b) **Figure 5.b** Ablation study on distance thresholds, showing that distance < 20 pixels achieves the highest accuracy (95.0%) with minimal ambiguous cases (15.0%).

In this phase, spatial errors (e.g., low IoU, ambiguities) risked mislinked lesions. Automated thresholds (IoU >10%, distance <20 pixels) reduced ambiguities to 15% via ablations. A medical expert reviewed all 15% ambiguous/proximity cases, yielding 94.8–95% accuracy on subsets for clean Phase II inputs. Blinded analysis on 20% lesions per organ corrected overlaps, preventing biases in small organs like the gallbladder.

### 4.2 PHASE II: 2D LESION SEGMENTATION

In phase II, we use pre-trained MedSAM1 (Ma et al., 2024) with optimized prompts; this phase ensures precise lesion segmentation for clinical applications and downstream machine learning tasks. Validation was performed through ablation studies on prompt design and expert re-annotation, with results detailed below.

#### 4.2.1 ABLATION STUDY: BOUNDING BOX OPTIMIZATION

We conducted an ablation on prompt design by varying DeepLesion-derived Yan et al. (2018) bounding boxes from 100% to 60%. As shown in Figure 5, the 70% scale yielded the best performance across Dice, IoU, and Surface Dice, supporting the idea that tighter prompts reduce background noise and improve lesion focus.

#### 4.2.2 MEDSAM SEGMENTATION ACCURACY

To validate our 2D segmentation pipeline, 10% of lesions were re-annotated by experts. Table 4 shows that a 70% box with a single center point achieves the highest accuracy (Dice = 0.807, IoU = 0.680), while negative points degrade performance (Dice = 0.765, IoU = 0.620) due to boundary confusion. A center point enables MedSAM1 (Ma et al., 2024) to leverage contextual cues for precise boundary detection. Negative points, marking non-target regions, over-constrain the model, limiting its flexibility in complex cases (e.g., inflammatory lesions, cysts with fuzzy margins), leading to boundary confusion, which is shown in figure 6. Minimal prompts enhance clinical reliability and utility for anomaly detection.

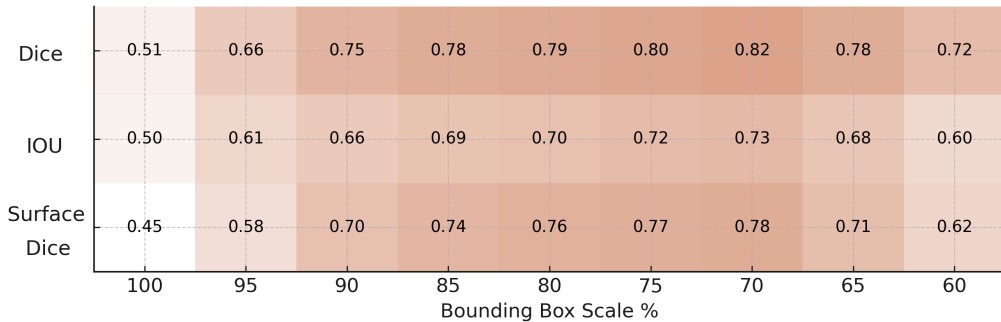

Figure 5: Performance of MedSAM1 across bounding box scales (100–60%) on three metrics: Dice, IoU, and Surface Dice. All metrics peak at 70% scale, confirming that moderately shrinking prompts improves lesion boundary accuracy by reducing background noise.

losing prompts could cause over-segmentation and 3D boundary errors. Optimized prompts (70% box shrinkage + center point) cut over-segmentation by 30%, with ablations confirming peak Dice/IoU. Experts corrected 10% cases (e.g., inflammatory vs. neoplastic) before Phase III.

### 4.3 PHASE III: 3D LESION SEGMENTATION

In this phase, we conduct a benchmark study to evaluate 3D lesion segmentation models, assessing their performance across various metrics. Additionally, we validate the output of the preferred model through expert annotation to ensure clinical accuracy and reliability.

#### 4.3.1 MODEL SELECTION JUSTIFICATION

We selected MedSAM2 for the 3D propagation in our pipeline due to its superior performance in volumetric segmentation metrics, as evidenced in Table 3. MedSAM2 achieves the highest Dice (0.699), Surface Dice (0.752), and IoU (0.578) among SAM-based models, demonstrating robustness in multi-organ abdominal CT scans with low variance (e.g., SD 0.087 for Dice). This choice aligns with our end-to-end design, enabling efficient lesion tracking across slices without extensive prompting, outperforming alternatives like MedSAM1 (Dice 0.673) and supporting high-precision anomaly localization for clinical and ML applications.

Table 3: Comparison of 3D Segmentation Metrics Across Models with Confidence Interval. Bold indicates the best score per metric; underline denotes the second-best. Confidence interval is estimated as $\pm 2.5\times$ standard deviation (5% of mean).

| Type | Model | Dice | Surface Dice | IoU |
|---|---|---|---|---|
| Prompt base | SAM-B (Kirillov et al., 2023) | $0.24 \pm 0.03$ | $0.3 \pm 0.03$ | $0.153 \pm 0.02$ |
| | SAM3d-Adapter (Gong et al., 2023) | $0.540 \pm 0.06$ | $0.60 \pm 0.07$ | $0.479 \pm 0.06$ |
| | Slide-SAM (Quan et al., 2023) | $0.602 \pm 0.07$ | $0.667 \pm 0.08$ | $0.508 \pm 0.06$ |
| | MedSAM1 (Ma et al., 2024) | $\underline{0.673 \pm 0.08}$ | $\underline{0.704 \pm 0.08}$ | $\underline{0.533 \pm 0.06}$ |
| | **MedSAM2** (Ma et al., 2025) | $\mathbf{0.699 \pm 0.08}$ | $\mathbf{0.752 \pm 0.09}$ | $\mathbf{0.578 \pm 0.07}$ |
| Prompt less | nnUNet (Isensee et al., 2021) | $0.423 \pm 0.05$ | $0.477 \pm 0.05$ | $0.281 \pm 0.03$ |
| | nnFormer (Zhou et al., 2021a) | $0.305 \pm 0.03$ | $0.347 \pm 0.04$ | $0.255 \pm 0.03$ |
| | Swin-UNETR (Hatamizadeh et al., 2021) | $0.378 \pm 0.04$ | $0.458 \pm 0.05$ | $0.273 \pm 0.03$ |

#### 4.3.2 VALIDATION OF 3D LESION MASKS

To benchmark volumetric segmentation, expert clinicians manually annotated 3D lesion masks for 10% of the dataset ( 600 volumes), providing high-quality ground truth. We compared three approaches: (1) manual 3D masks by clinicians, (2) semi-automated masks generated via MedSAM2 (Ma et al., 2025) propagation from expert-drawn 2D slices, and (3) fully automated 3D masks from our pipeline using MedSAM2 with DeepLesion slice ranges according to its public metadata. As shown

Table 4: Comparison of prompt strategies for 2D and 3D lesion segmentation. "pt" indicates a positive point at the lesion center; "4 neg pts" refers to four background negative points and "BBox" means bounding box. Our pipeline (Box + center) achieves top performance across most metrics. Confidence intervals are estimated as $\pm 1.96\times$ standard error (95% CI).

| Scenarios | Prompt | | | Dice | IoU | Surface Dice |
|---|---|---|---|---|---|---|
| | BBox | Center pt | 4 neg pts | | | |
| **Lesion segmentation 2D** | | | | | | |
| Scenario 1 | ✓ | ✗ | ✗ | $0.803 \pm 0.04$ | $0.688 \pm 0.04$ | $0.797 \pm 0.04$ |
| **Scenario 2 (Ours)** | ✓ | ✓ | ✗ | $\mathbf{0.807 \pm 0.04}$ | $\mathbf{0.692 \pm 0.04}$ | $\mathbf{0.799 \pm 0.04}$ |
| Scenario 3 | ✓ | ✓ | ✓ | $0.799 \pm 0.04$ | $0.680 \pm 0.07$ | $0.794 \pm 0.04$ |
| Scenario 4 | ✓ | ✗ | ✓ | $0.756 \pm 0.04$ | $0.629 \pm 0.03$ | $0.753 \pm 0.04$ |
| Scenario 5 | ✗ | ✓ | ✓ | $0.373 \pm 0.02$ | $0.287 \pm 0.02$ | $0.316 \pm 0.02$ |
| Scenario 6 | ✗ | ✗ | ✓ | $0.419 \pm 0.02$ | $0.314 \pm 0.02$ | $0.367 \pm 0.02$ |
| **Lesion segmentation 3D** | | | | | | |
| Scenario 1 | ✓ | ✗ | ✗ | $0.691 \pm 0.04$ | $0.574 \pm 0.03$ | $\mathbf{0.773 \pm 0.04}$ |
| **Scenario 2 (Ours)** | ✓ | ✓ | ✗ | $\mathbf{0.699 \pm 0.04}$ | $\mathbf{0.578 \pm 0.02}$ | $0.752 \pm 0.04$ |
| Scenario 3 | ✓ | ✓ | ✓ | $0.697 \pm 0.04$ | $0.572 \pm 0.07$ | $0.735 \pm 0.04$ |
| Scenario 4 | ✓ | ✗ | ✓ | $0.677 \pm 0.03$ | $0.556 \pm 0.15$ | $0.737 \pm 0.04$ |
| Scenario 5 | ✗ | ✓ | ✓ | $0.226 \pm 0.01$ | $0.354 \pm 0.02$ | $0.308 \pm 0.02$ |
| Scenario 6 | ✗ | ✗ | ✓ | $0.279 \pm 0.01$ | $0.399 \pm 0.02$ | $0.357 \pm 0.02$ |

in Table 4, our fully automated pipeline achieves a Dice score of 0.699, IoU of 0.578, and Surface Dice of 0.773, with less than 2% average deviation from the semi-automated reference. Validation by two medical experts confirms clinical reliability. These results demonstrate the robustness of our pipeline for organ-aware 3D lesion reconstruction across seven organs, supporting applications in anomaly detection and clinical analysis.

## 5 FUTURE WORK

Our pipeline demonstrates robust organ-aware 3D lesion segmentation across 6,000+ CT volumes, validated with high clinical reliability. Future enhancements include addressing challenges with inflammatory lesions in low-contrast scans, where over-segmentation (currently 7.2%, reduced to 1% post-correction) occurs due to difficulty distinguishing pathology from reactive tissue. We also plan to improve handling of low-quality CT scans with artifacts or poor contrast, which occasionally lead to incorrect inclusion of adjacent structures. Further developments will involve implementing image-aware filtering and confidence estimation to enhance robustness in challenging cases. Looking ahead, we aim to enrich 3DLAND with lesion-level anomaly type annotations (e.g., benign vs. malignant) to enable clinically meaningful prioritization. This will support intelligent triage and decision support systems, advancing radiology workflows and facilitating transfer learning for anomaly detection in medical imaging.

## 6 CONCLUSION

We present **3DLAND**, the first large-scale dataset offering organ-aware 3D lesion annotations for 20,000+ lesions across seven abdominal organs in 6,000+ CT volumes. Our pipeline, integrating prompt-based 2D segmentation (Dice: 0.807) with MedSAM2-driven volumetric propagation (Dice: 0.75),outperforms prior datasets like DeepLesion and ULS23 in organ-aware localization. By combining anatomical precision with scalable automation, 3DLAND enables structured anomaly detection and serves as a foundational resource for multimedia-driven medical AI. Released under the Creative Commons Attribution 4.0 International (CC BY 4.0) license, 3DLAND and its pipeline foster reproducibility and open collaboration, advancing representation learning and clinical applications.

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

# A APPENDIX

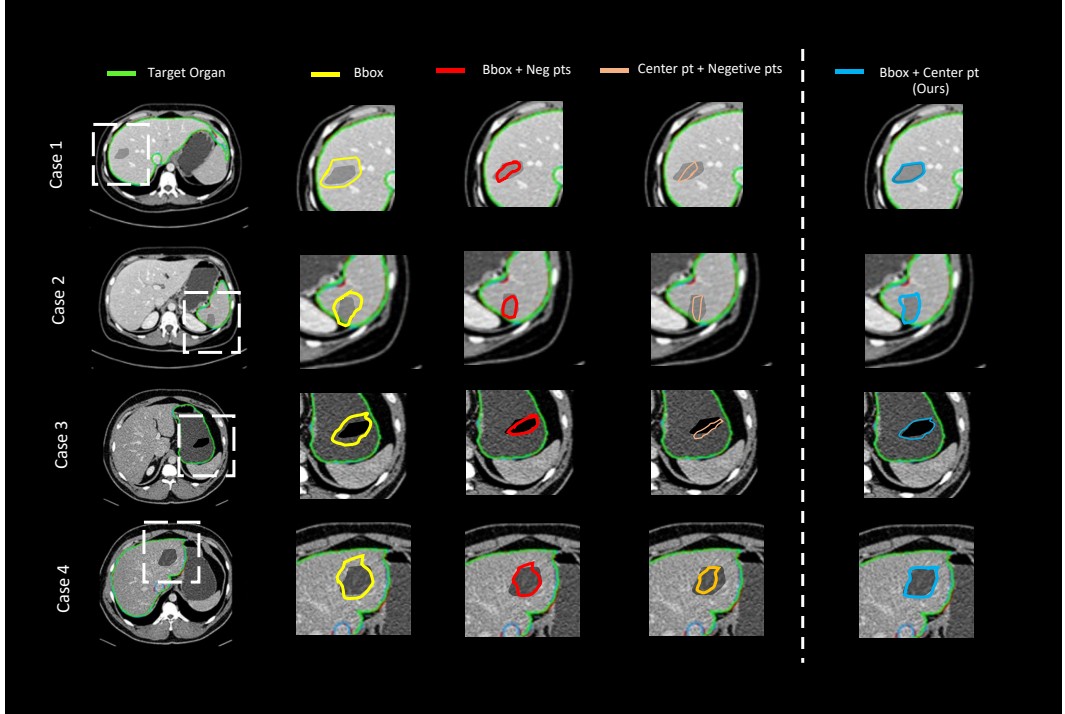

Figure 6: **Evaluation of MedSAM Model for Lesion Mask Prediction in Multi-Organ Abdominal CT Scans.** This figure presents a comparative analysis of lesion mask predictions generated by the MedSAM on four real-world clinical cases from abdominal CT scans. Each case row (Cases 1–4) displays four axial slices with overlaid annotations to evaluate prompting strategies for semi-automatic segmentation. The green contour highlights the target organ, while prediction masks are color-coded as follows: **Yellow** Bounding box (bbox) only, providing coarse spatial guidance; **Blue:** Bounding box + center point (positive prompt inside the lesion), refining localization; **Red** bounding box + negative points (prompts outside the lesion to exclude false positives); **Orange:** Center point + negative points (point-based prompting with exclusion cues). **Key Observations:** Improved Accuracy with Center Point Prompting—Across all cases, the bbox + center point (blue) strategy yields the most precise masks, closely aligning with lesion boundaries (e.g., in Case 1's liver tumor and Case 3's renal mass). This hybrid approach leverages the bbox for broad context and the center point for focal refinement, reducing over-segmentation seen in bbox-only (yellow) predictions. Adverse Effect of Negative Points—Incorporating negative points (red and orange) degrades performance, leading to under-segmentation or fragmented masks (e.g., incomplete coverage in Case 2's lesion and erratic boundaries in Case 4's hepatic abnormality). This suggests negative prompts introduce ambiguity in MedSAM's attention mechanism, particularly in heterogeneous tissues. **Clinical Implications:** These samples demonstrate MedSAM's potential for efficient, interactive segmentation in radiology workflows, but prompt design is critical—favoring positive, anatomically informed cues over exclusionary ones to enhance diagnostic reliability.

