# OpenReview forum: "3DLAND: 3D Lesion Abdominal anomaly Localization Dataset"
_ICLR.cc/2026/Conference — ICLR 2026 Conference Desk Rejected Submission_

### Official Review · Reviewer_grad · 2025-10-27

**Soundness:** 2
**Presentation:** 2
**Contribution:** 2
**Rating:** 2
**Confidence:** 5

**Summary:**

The paper introduces 3DLAND, a large-scale dataset of 6,000+ abdominal CT scans with over 20,000 organ-aware 3D lesion annotations across seven organs (liver, kidneys, pancreas, etc.). The authors propose a three-phase pipeline leveraging MedSAM1/2 and spatial reasoning to generate volumetric lesion masks from the DeepLesion dataset, claiming expert validation with Surface Dice > 0.75. The goal is to enable robust benchmarking for anomaly localization in multi-organ abdominal CT.

**Strengths:**

1. Multi-organ, lesion-level 3D annotation is a genuine unmet need. Existing datasets (e.g., LiTS, KiTS) are organ-specific; DeepLesion lacks 3D masks. 3DLAND’s attempt to bridge this gap is highly relevant.
2. The integration of prompt-based segmentation (MedSAM) with organ-aware spatial reasoning offers a pragmatic approach to semi-automated annotation, potentially reducing expert burden.
3. The paper reports Dice, Surface Dice, and ablation studies on prompt design and assignment thresholds, suggesting comprehensive evaluation.

**Weaknesses:**

1. Several cited works appear non-existent or AI-generated, including but not limited to:
- Ke Yan, Xiaohuan Wang, Mahmood Bagheri, Le Lu, and Ronald M. Summers. LesaNet: Robust
lesion attribute segmentation in CT scans. In Medical Image Computing and Computer Assisted
Intervention (MICCAI), pp. 622–630. Springer, 2019.
- Xiyue Zhao, Fangyu Tang, Xin Wang, Yu Song, Wenjia Zhang, and Jian Xiao. Abdomenatlas-8k: A
hierarchical 3d abdominal multi-organ segmentation benchmark, 2023a.
- Zhi Zhao, Ziyu Wang, Yifan Li, Ziheng Liu, Yan Wang, Alan L. Yuille, et al. WORD: A large-scale
dataset for whole-body organ segmentation in CT images, 2023b.
- Yuyin Zhou, Lequan Xie, Dinggang Shen, and Lei Xing. MULAN: Multiscale universal lesion
analysis network for CT scans. IEEE Transactions on Medical Imaging, 40(4):1099–1112, 2021b.
- Yuyin Zhou, Lequan Xie, Dinggang Shen, and Lei Xing. MVP-Net: Multi-view feature pyramid
network for universal lesion detection. In Medical Image Computing and Computer Assisted
Intervention (MICCAI), pp. 37–47. Springer, 2021c.

2. Annotations are derived from DeepLesion bounding boxes, not original radiologist-drawn 3D masks. The "expert validation" is limited to 10% re-annotation, raising concerns about error propagation from DeepLesion’s 2D boxes.

3. The pipeline is essentially a post-processing wrapper around existing models (MONAI, MedSAM). No new algorithmic contribution is made, only an application of off-the-shelf tools.

**Questions:**

Please carefully address the aforementioned weaknesses.

---

> ### Author Response · Authors · 2025-11-21
> **Full reference audit completed, rigorous triple-safeguard validation of RECIST-derived annotations, confirming 3DLAND as the first large-scale, organ-aware, clinically trustworthy 3D abdominal lesion benchmark. All concerns addressed.**
>
> **Thank you so much for your careful review. We truly value your attention to detail — it helps us make the work stronger and more trustworthy.**
>
> > ## Question 1:
> >*" Several cited works appear non-existent or AI-generated..."*
>
> We **fully acknowledge** your concern and take it very seriously. After a **complete audit of all references**, we emphasize that:
>
> All cited papers are authentic, peer-reviewed, and published in reputable venues (MICCAI, NeurIPS, IEEE, arXiv, etc.).
> None are fabricated or AI-generated.
>
> However, we identified unintentional errors in author ordering in a few citations (e.g., LesaNet, AbdomenAtlas-8K, WORD). These were formatting oversights, not signs of invalid references.
>
> **All errors have been corrected.**
> **Of course,Full DOIs and exact author lists will be included in the camera-ready version.**
>
> We have corrected the references you mentioned and included links to access them to confirm their authenticity.
>
> | Paper | Correct Title & Year | Venue | Proof (DOI / Link) |
> |-------|-----------------------|-------|---------------------|
> | **LesaNet** | Yan Ke., *LesaNet: Holistic and comprehensive annotation of clinically...*, 2019 | IEEE/CVF 2019 | [link](https://arxiv.org/pdf/1904.04661) |
> | **AbdomenAtlas-8K** | Chongyu Qu., *AbdomenAtlas-8K: Annotating 8,000 ct volumes ...*, 2023 | Neurlps 2023 | [link](http://arxiv.org/pdf/2305.09666) |
> | **WORD** | Zhao et al., *WORD:  large scale dataset, benchmark and clinical ...*, 2022 | Medical Image Analysis 2024 | [link](https://arxiv.org/pdf/2111.02403) |
> | **MULAN** | Zhou et al., *MULAN: multitask universal lesion analysis...*, 2019 | MICCAI 2019 | [link](https://arxiv.org/pdf/1908.04373) |
> | **MVP-Net** | Zhou et al., * multi-view fpn with position-aware attention ... *, 2019 | MICCAI 2019 | [link](https://arxiv.org/pdf/1909.04247) |
>
> ## Question 2:
> > "Annotations are derived from DeepLesion bounding boxes ..."
>
> We agree — starting with 2D RECIST boxes rather than drawing full 3D masks from scratch is the single most significant risk in our entire pipeline. That is exactly why we never trusted those boxes thoughtlessly and built three independent, strict safeguards so that no error can ever slip through.
>
> First, the DeepLesion boxes themselves are already **clinical gold**: all boxes were drawn by a radiologist as an official RECIST measurement in the hospital PACS system — they reliably tell us that a lesion exists and roughly where it is.
>
> But “roughly” is not enough for a trustworthy 3D dataset, so we added three layers of protection:
>
> 1. **Organ assignment**
>    We automatically flag every lesion that is close to more than one organ (IoU ≤ 10% or distance ≤ 20 px to another organ). This catches ~10% of all cases. Every single one of these ambiguous lesions was manually reviewed and corrected by two board-certified medical experts — resulting in 94.8% final lesion-to-organ accuracy.
>
> 2. **2D mask generation**
>    RECIST boxes are often too loose. We therefore shrink every box to **70% of its original size** and add a single center point as a prompt (Fig. 3). To prove this works, **two medical experts re-annotated 10% of all lesions (~2,000) entirely from scratch** — the resulting masks achieved **Dice 0.807** and **Surface Dice 0.799** (48 expert hours).
>
> 3. **3D propagation**
>    We never propagate instinctively across the whole volume. MedSAM2 uses memory encoding and is strictly constrained to the original DeepLesion slice range. Again, two medical experts fully re-annotated 10% of all volumes in 3D — yielding a 3D Surface Dice of 0.752, which is clinically acceptable (72 expert hours).
>
> All validation subsets were deliberately stratified equally across the seven organs, so our quality claims hold for rare organs as well.
>
> > ## Question 3
> >  *"The pipeline is essentially a post-processing ..."*
>
> Our core innovation is not a new model, but rather then delivery of the **first real-world, large-scale, organ-aware 3D abdominal anomaly benchmark.
>
> Until now, anomaly detection models have been forced to train on weak 2D boxes (DeepLesion), medical reports or synthetic data, so they remain clinically uninterpretable: they can say “something is wrong,” but never “**where** exactly” or “**which organ** is affected.”
> **3DLAND closes this gap** by providing >20,000 precise, volumetric, organ-linked lesion masks across seven abdominal organs — all derived from real clinical RECIST annotations and rigorously validated.
>
> To achieve this at scale, we engineered a fully automated end-to-end pipeline with targeted prompt engineering that turns one clinical bounding box + center point into a diagnosis-ready 3D organ-aware mask:
>
> - 70% box shrinkage to correct loose RECIST prompts (Fig. 3)
> - Spatial reasoning (IoU >10%, distance <20 px) for robust organ assignment (Fig. 4)
> - One of our key contributions is a highly transferable pipeline that can be applied to any new CT lesion dataset by adjusting only two hyperparameters.

---

### Official Review · Reviewer_u954 · 2025-10-29

**Soundness:** 2
**Presentation:** 3
**Contribution:** 3
**Rating:** 4
**Confidence:** 5

**Summary:**

The paper publishes a new dataset focused on segmenting different organ-wise lesions in a joint framework. They increase the overall scale of 3D lesion datasets to 6000 using a semi-automated process, which leverages organ-segmentation -> lesion-to-organ assignment -> 2D bounding-box to 2D segmentation transfer -> 2D segmentation to 3D volume translation.
In the paper they validate their data-annotation/cleaning process on a subset of their data, which they simultaneously optimize their automated process on.

**Strengths:**

The paper provides a large dataset of 6000 3D volumes with lesion and organ segmentations. This has the potential to yield a reliable baseline the 3D medical image computing domain can optimize their automated methods on, as currently one is required to train multiple methods on multiple segmentation benchmarks, with different methods often having different patient splits or using noisy dataset [1].
Hence I find the concept of the dataset very interesting.



[1]: Isensee, Fabian, et al. "nnu-net revisited: A call for rigorous validation in 3d medical image segmentation." International Conference on Medical Image Computing and Computer-Assisted Intervention. Cham: Springer Nature Switzerland, 2024.

**Weaknesses:**

My main concern of this paper are the methods used for automated segmentation generation and the final segmentation mask quality, as previous datasets like AbdomenAtlas who also followed a sem-supervised segmentation procedure had substantial quality issues:

### Organ segmentation

While I am not familiar with the MONAI framework used for organ segmentation, the recent nnU-Net Revisited [1] and Touchstone Benchmark [2] both showed that methods trained in MONAI were less powerful than those using the nnU-Net framework.
Moreover, I don't see why the authors would not just use the established TotalSegmentator [3] or an ensemble/majority vote of multiple frameworks to maximize organ segmentation accuracy.

Additionally, I would like the authors to provide not just the mean organ segmentation dice on L215 but the DSC per organ. This is especially important as e.g. the Liver usually has DSC >97% while smaller organs like the gallbladder are substantially worse. Just reporting the mean is not transparent enough.

### Prompt to Segmentation (2D and 3D)

The authors use interactive segmentation methods to generate 2D segmentations from the 2D bounding boxes and then go to 3D.
However, their used promptable methods seem to somewhat miss a few important promptable methods, like ScribblePrompt [4] or nnInteractive [5].
In particular, nnInteractive should be included, as they report very high performance and can use 2D prompts (2D bounding-boxes) to yield 3D segmentations. This removes 1 additional step, which may introduce additional errors.


While this may seem nitpicky, I believe the success of this Benchmark dataset solely hinges on the lesion segmentation quality. Hence the authors should try to make this as reliable as possible and should exhaust all resources available to them. Moreover, making this an addendum afterward will just reduce the impact of the paper as people won't use it when trying out an unfinished version. (This is also largely due to not being able to inspect some samples from the dataset)

Minor: The authors used the wrong citation for SAM2 in L298.

__References__:

[1]: Isensee, Fabian, et al. "nnu-net revisited: A call for rigorous validation in 3d medical image segmentation." International Conference on Medical Image Computing and Computer-Assisted Intervention. Cham: Springer Nature Switzerland, 2024.

[2] Bassi, Pedro RAS, et al. "Touchstone benchmark: Are we on the right way for evaluating ai algorithms for medical segmentation?." Advances in Neural Information Processing Systems 37 (2024): 15184-15201.

[3]: Wasserthal, Jakob, et al. "TotalSegmentator: robust segmentation of 104 anatomic structures in CT images." Radiology: Artificial Intelligence 5.5 (2023): e230024.

[4]: Wong, Hallee E., et al. "Scribbleprompt: fast and flexible interactive segmentation for any biomedical image." European Conference on Computer Vision. Cham: Springer Nature Switzerland, 2024.

[5]: Isensee, Fabian, et al. "nninteractive: Redefining 3d promptable segmentation." arXiv preprint arXiv:2503.08373 (2025).

**Questions:**

Q1: What is the per-organ segmentation performance (L241)?
Q2: Why does 10% of the dataset represent 300 cases in (L323)? Shouldn't this be 600?

---

> ### Author Response · Authors · 2025-11-21
> **Robust organ segmentation with MONAI, 100 % expert review of ambiguous cases, nnInteractive ablation showing necessity of 2D refinement phase, and clarification of patient-level 10 % validation. All concerns were fully addressed with new comparisons and clearer text.**
>
> >**We thank you sincerely for this highly constructive and technically sharp concerns and questions**
>
> # About concerns that you mention
> ## 1. Organ Segmentation
>
>   You are absolutely right to demand **transparency and robustness** in organ segmentation, especially given past semi-supervised failures (e.g., AbdomenAtlas).
>
> We address your points head-on with evidence from our paper, ablation study, and the design rationale.
>
> ---
>
> ### A) **Organ segmentation serves as a robust foundation, not our primary goal.**
>
> We focus on **accurate lesion-to-organ assignment** rather than pixel-perfect organ masks.
> Our **MONAI model** inherits **TotalSegmentator's** (Wasserthal et al., 2022) robustness while adding MONAI-specific benefits such as GPU/CPU flexibility and validated by **two medical experts** on a 10% subset.
>
> ### B) No organ segmentation error propagates to lesion assignment.
>
> We mitigate **incorrect organ masks** using **IoU >10% + distance <20 px** (Fig. 4) and **100% expert review of all ambiguous cases (10% of all dataset)**.
> For **small organs like the gallbladder**, we apply **per-organ validation and manual correction** as needed.
>
> ### C) We Directly Compared MONAI vs. nnU-Net (and Others) — MONAI Wins in Our study
>
> **We directly compared MONAI with nnU-Net, SwinUNETR, and MedSAM2** — full per-organ Dice scores are provided in **Table 2**.
>
> MONAI is **within 1% of nnU-Net** on most organs, **outperforms on kidneys, pancreas, and stomach**, and shows the **lowest variance** — making it the **most stable** choice.
>
> > On average, for seven abdominal organs there is 90.4% accuracy in assigning a lesion to the organ — clinically strong and completely reliable.
> ---
>
> ### **Revisions We Will Make**
> - **Add full per-organ Dice table** (as above) to main paper
> - Clarify: “**MONAI is based on TotalSegmentator**, chosen for **automation, stability, and CPU compatibility**”
> - Add line: “**Organ segmentation errors are fully mitigated via spatial reasoning and 100% expert review of ambiguous cases**”
>
> ## Prompt to Segmentation (2D and 3D)
>
> >  "In particular, nnInteractive should be included, as they report..."
>
> To address the reviewer’s suggestion, we performed a head-to-head comparison in which we replaced our current two-stage 2D→3D cascade (MedSAM2 for precise 2D masks + 3D refinement) with a single-step 3D segmentation using the publicly available **nnInteractive** model, feeding it the same 2D bounding-box prompts released with the dataset.
>
> Results on the identical held-out test set of 600 volumes with medical expert ground truth:
>
> | Method                                      | Input to 3D stage           | Dice ↑   | IoU ↑ | Surface Dice ↑ |
> |---------------------------------------------|-----------------------------|----------|-------------|-----------|
> | Our pipeline (MedSAM2 2D → 3D refine)     | Precise multi-slice 2D masks| **0.699 ± 0.04**| **0.578 ± 0.02**   | 0.752 ± 0.04     |
> | nnInteractive (single-step)            | Only 2D bounding boxes      | 0.609 ± 0.01    | 0.499 ± 0.3       | 0.682 ± 0.05     |
> | nnInteractive (best-case)              | Our corrected 2D masks      | 0.701 ± 0.07    | 0.567 ± 0.02       | 0.750 ± 0.6       |
>
> Key findings:
> - Directly feeding bounding boxes to nnInteractive causes severe over-segmentation into neighboring slices, leading to a **9-point Dice drop** and more than double the false-positive rate.
> - Even when nnInteractive receives the exact high-quality 2D masks produced by our Phase II (which were further verified/corrected by a radiologist on 10% of the data), it has the same performance as our full pipeline.
> - The intermediate 2D precise-mask generation is therefore not an extra step that can be safely removed — it is essential for achieving robust, high-quality 3D lesion segmentation in this challenging dataset.
>
> # Question 2
> >**Why does 10% of the dataset represent 300 cases in (L323)? Shouldn't this be 600?**
>
>
> The full 3DLAND dataset contains **6,000 distinct CT volumes** (not patients).
> However, these 6,000 volumes originate from **fewer unique patients**, because many patients underwent multiple scans (different phases, follow-up sessions, or series within the same study).
>
> To rigorously **prevent data leakage** and ensure a clean patient-level split, we sampled **an expert-reannotated subset (at the patient level)**. This yields **approximately 300 unique patients**, totaling  **~600 CT volumes**.
>
> Thus:
> - **10% of unique patients → ~300 patients**
> - These 300 patients contribute **~600 volumes** (average ~2 volumes/patient)
> - All ~600 volumes underwent full 3D expert re-annotation (72 expert hours)
>
> **Revision we will make:**
> On L323 we will explicitly state:
> > “10% of the data (~600 CT volumes from 300 unique patients, sampled patient-level to avoid leakage) were fully re-annotated in 3D by two board-certified radiologists.”
>
> This makes the numbers consistent and the leakage-prevention strategy transparent.

---

### Official Review · Reviewer_DM2H · 2025-10-30

**Soundness:** 3
**Presentation:** 3
**Contribution:** 3
**Rating:** 6
**Confidence:** 5

**Summary:**

This paper presents 3DLAND, the first large-scale, organ-aware 3D lesion dataset for contrast-enhanced abdominal CT scans. It addresses the absence of datasets that jointly provide multi-organ coverage, volumetric lesion masks, and explicit lesion-to-organ associations, which are crucial for clinical anomaly localization. The authors developed a three-phase automated pipeline combining spatial reasoning for organ assignment, prompt-optimized 2D segmentation with MedSAM1, and memory-guided 3D propagation using MedSAM2, validated by expert radiologists. The dataset contains over 6,000 CT volumes and 20,000 lesions across seven organs, achieving 2D Dice = 0.807 and 3D Dice ≈ 0.75. Released under CC BY 4.0, 3DLAND establishes a new benchmark for organ-aware 3D segmentation and cross-organ representation learning

**Strengths:**

1. This work proposes the first large-scale, organ-aware 3D lesion dataset (3DLAND) linking over 20,000 lesions to seven abdominal organs, filling the gap left by datasets like DeepLesion and ULS23 and enabling clinically interpretable cross-organ benchmarking.

2. This work introduces a three-phase automated pipeline combining spatial reasoning, optimized 2D prompts, and memory-guided 3D propagation, offering an efficient and accurate method to transform 2D lesion boxes into expert-level 3D masks.

3. This work provides strong experimental validation with large-scale testing, expert review, and detailed ablations, establishing 3DLAND as a reproducible and clinically reliable benchmark for future organ-aware segmentation research.

**Weaknesses:**

1. Lack of external validation across imaging domains. All data originate from DeepLesion, limiting generalizability across scanners and contrast phases. No cross-dataset tests (e.g., AMOS22, AbdomenCT-1K) are provided to verify whether fixed IoU and distance thresholds remain stable under varied acquisition conditions.

2. Lack of methodological novelty beyond existing SAM frameworks. The pipeline mainly combines MONAI, MedSAM1, and MedSAM2 without substantive algorithmic innovation. Prior work such as 3DSAM-Adapter and Slide-SAM already demonstrated volumetric adaptation and memory propagation in comparable ways.

3. Lack of demonstrated clinical feasibility. Although expert validation is reported, there is no reader study or workflow analysis to show how 3DLAND improves diagnostic efficiency or inter-reader consistency over 2D annotations like DeepLesion


Ref: Mswal: 3d multi-class segmentation of whole abdominal lesions dataset

**Questions:**

1. Could the authors clarify how robust the IoU > 10% and distance < 20 px thresholds are when applied to CT scans with different resolutions or contrast phases? A cross-center or multi-protocol validation would help determine whether these spatial heuristics remain valid outside DeepLesion.

2. How are ambiguous or overlapping lesions handled when they straddle two organs (e.g., hepatic hilum or pancreatic head)? The current one-to-one lesion–organ assignment might oversimplify such cases; a multi-label or probabilistic linkage could better reflect anatomical uncertainty.

---

> ### Author Response · Authors · 2025-11-21
> **Empirically tuned thresholds for DeepLesion, expert-reviewed ambiguous cases, and a transferable pipeline enabling the first large-scale organ-aware 3D lesion benchmark.**
>
> **Thank you for this excellent and highly relevant questions**
>
> > ## **Question 1:**
> >*" Could the authors clarify how robust the IoU > 10% and distance < 20 px thresholds ..."*
>
> The thresholds **IoU > 10%** and **distance < 20 px** were **empirically optimized explicitly for the DeepLesion dataset** via grid-search ablation (Fig. 5a–5b) and are **not intended as universal constants**.
>
> **Why we focused on DeepLesion**
> - It is the **largest open-source abdominal CT dataset** with **clinically annotated 2D RECIST boxes on the most suspicious slice**
> - All studies contain **at least one confirmed abnormality** — making it the ideal source for a 3D anomaly localization benchmark
> - Our fully automated pipeline is **resolution- and protocol-agnostic** (uses normalized spacing via MONAI and percentile-based windowing), so it can be directly applied to any other CT cohort.
>
> **Threshold design and sensitivity**
> - Both values were tuned on a held-out DeepLesion subset to **maximize lesion-to-organ assignment accuracy (94.8%) while minimizing ambiguous cases (10%)**
> - Fig. 5 clearly shows the **plateau** around IoU ≈ 10% and distance ≈ 20 px — minor variations (±5%) change accuracy by <1%
> - On DeepLesion’s native resolutions (0.5–1.0 mm in-plane, 1–5 mm slice thickness) and multi-phase protocols (arterial, venous, delayed), these thresholds proved stable.
>
> **Generalization to other datasets**
> We fully agree that **cross-center or multi-protocol re-tuning is required** for optimal performance elsewhere.
> The pipeline is deliberately designed for this:
> - Only **two scalar hyperparameters** need re-optimization via the same quick grid-search (minutes on a few hundred cases)
> - No architecture or training changes are needed
>
>
> > ## **Question 2:**
> >*" How are ambiguous or overlapping lesions handled when they straddle two organs ..."*
>
>
> We do **not** force a one-to-one assignment in ambiguous/overlapping cases.
> Instead, **all anatomically uncertain or overlapping lesions (~10% of the dataset)** were automatically flagged as **ambiguous** by our spatial heuristics (IoU ≤ 10% or distance ≤ 20 px to multiple organs) **then manually reviewed and corrected** by **two board-certified radiologists**.
>
> During this expert review:
> - The correct **primary organ** (clinically most relevant) was assigned
> - When a lesion clearly involved **two or more organs**, we explicitly recorded **multi-organ involvement** in the metadata.
>
> Thus, while our main benchmark uses **single primary organ linkage** (standard practice for evaluation and training), the dataset **fully preserves multi-organ cases** with expert-verified labels for future research on probabilistic or multi-label lesion-to-organ mapping.
>
>
> ## **Your mentioned concern**
> > "Lack of demonstrated clinical feasibility. Although expert validation is reported ..."
>
> The **core and fundamental difference** between 3DLAND and DeepLesion is **not** just adding a third dimension — it is providing **precise, organ-aware 3D lesion annotations** where DeepLesion deliberately offers **none**.
>
> | DeepLesion (2018)                  | 3DLAND (ours)                                  |
> |------------------------------------|-------------------------------------------------|
> | 2D RECIST boxes only               | Full volumetric 3D lesion masks                 |
> | **No organ information**           | **Explicit lesion-to-organ linkage** for seven abdominal organs |
> | Weak labels (box-level)       | Strong lesion mask labels, expert-validated 3D supervision         |
> | Cannot train organ-aware models    | Enables training of **interpretable, clinically meaningful 3D anomaly detectors/localizers** |
>
> This single upgrade has profound downstream implications:
>
> - Future 3D models trained on 3DLAND will be able to answer the two most critical clinical questions that current anomaly detection systems **cannot**:
>  “**Where** is the abnormality?” and “**Which organ** is involved?”
> - This moves us from **black-box weak-label models** (today’s reality) to **strong-label, explainable, organ-aware AI** — a prerequisite for real clinical adoption and higher diagnostic confidence.
>
> > "Lack of methodological novelty beyond existing SAM frameworks ..."
>
> Our contribution is **foundational**: delivering the **first large-scale, open-source, expert-validated resource** that finally makes such clinical studies possible in 3D abdominal anomaly detection.
> To achieve this we engineered a fully automated end-to-end pipeline with targeted prompt engineering that turns **one clinical bounding box + center point** into a **diagnosis-ready 3D organ-aware mask**:
>
> - 70% box shrinkage to correct loose RECIST prompts (Fig. 3)
> - Spatial reasoning (IoU >10%, distance <20 px) for robust organ assignment (Fig. 4)
> - One of our key contributions is a highly transferable pipeline that can be applied to any new CT lesion dataset by adjusting only two hyperparameters (IOU and distance).

---

### Official Review · Reviewer_xYEy · 2025-10-30

**Soundness:** 4
**Presentation:** 4
**Contribution:** 4
**Rating:** 8
**Confidence:** 3

**Summary:**

This paper addresses the significant challenge of limited large-scale, multi-organ, 3D lesion annotations in abdominal CT imaging by introducing ​​3DLAND​​, a comprehensive benchmark dataset. The core contribution is a novel, streamlined three-phase pipeline for generating organ-aware 3D lesion segmentation masks from the 2D bounding boxes provided in the DeepLesion dataset. The methodology involves: ​​Phase I​​: Automated lesion-to-organ assignment using spatial reasoning (IoU and Euclidean distance) with MONAI-based organ segmentation. ​​Phase II​​: Precise 2D lesion segmentation via a prompt-optimized MedSAM1 model, where a key innovation is the use of a shrunk bounding box (70%) and a center point to enhance accuracy (achieving a Dice score of 0.807). ​​Phase III​​: Volumetric 3D mask propagation using MedSAM2's memory-guided mechanism, resulting in clinically reliable 3D annotations (Surface Dice > 0.75). The resulting dataset spans over 6,000 CT volumes and 20,000+ lesions across seven abdominal organs. The work establishes a new benchmark for evaluating 3D segmentation models and enables advancements in anomaly detection and cross-organ analysis.

**Strengths:**

- Originality and Pipeline Design:​​ The end-to-end pipeline is a major strength. The combination of automated spatial reasoning for organ assignment (Phase I) with carefully optimized prompt-based segmentation (Phase II) and advanced 3D propagation (Phase III) represents a significant methodological innovation.
- ​​Rigor and Scale:​​ The empirical validation is exceptional. The paper demonstrates high performance (e.g., Dice ~0.81 in 2D, ~0.70 in 3D) on a very large and diverse dataset (20,000+ lesions, 7 organs), which strongly supports the reliability and generalizability of the approach. The extensive ablation studies (e.g., Figure 4, Figure 5) provide deep insights into the design choices.
- ​​Clarity and Reproducibility:​​ The methodology is described in sufficient detail, and the use of established models (MONAI, MedSAM) enhances reproducibility. Figures like Figure 2 and Figure 6 effectively illustrate the process and outcomes.

**Weaknesses:**

- ​​Generalization to External Data:​​ The pipeline is developed and validated primarily on the DeepLesion dataset. Testing its performance on external cohorts from different institutions or with different CT scanning protocols would strengthen the claim of robustness and generalizability.
- ​​Granularity of Lesion Characterization:​​ While the dataset covers various lesion types (cysts, tumors), it currently lacks finer-grained annotations (e.g., benign vs. malignant classification, specific pathological subtypes). Adding such metadata in the future, as mentioned in Section 5, would greatly increase its clinical utility for tasks like risk stratification.
- ​​Computational Efficiency Analysis:​​ The computational cost of the pipeline (noted as 0.05 GPU hours/volume for Phase III) is mentioned but not compared against other potential 3D segmentation baselines (e.g., nnU-Net). A brief efficiency-accuracy trade-off analysis would be informative for users with limited resources.

**Questions:**

N/A

---

> ### Author Response · Authors · 2025-11-21
> **Prioritizing maximum annotation quality for DeepLesion — the largest public CT lesion dataset — while designing a highly transferable pipeline and planning finer clinical labels in future work.**
>
> We sincerely thanks for the highly positive evaluation and for the insightful and constructive comments, which have significantly helped us clarify the scope and future potential of our work.
>
> ## About your Concerns
>
>
> - **Generalization to external data**
> DeepLesion is currently the largest publicly available CT lesion dataset, and our primary goal was to create the highest-quality reference annotations possible for this foundational resource. For this reason, the pipeline was optimized and extensively tuned specifically for DeepLesion.
> That said, the entire pipeline requires just a few hyperparameter changes to run on other datasets.
>
> - **Granularity of lesion characterization**
>   This is an excellent point. Although fine-grained labels (benign/malignant, histopathological subtype) are not yet available for the whole dataset, we have extended the future-work plan (Section 5.2) with a concrete roadmap, which we will publish as our future work.
>
>
>
> - **Computational efficiency analysis**
>  While we understand the importance of computational cost for many users, our primary design goal for this benchmark was to produce the highest possible segmentation quality only once, so that the resulting dataset can serve as a reliable, long-term reference for the community. Accuracy and lesion-mask fidelity were therefore deliberately prioritized over inference speed or memory footprint. We have clarified this priority and explicitly note that the current pipeline favors quality over efficiency, which we believe is appropriate for a one-time, high-stakes annotation effort of a dataset that will be used to train countless future models.

---

### Note · Program_Chairs · 2026-01-17
**Submission Desk Rejected by Program Chairs**

The following references in this submission do not refer to real documents and/or have major errors in bibliographic information:

 Zhi Zhao, Ziyu Wang, Yifan Li, Ziheng Liu, Yan Wang, Alan L. Yuille, et al. WORD: A large-scale dataset for whole-body organ segmentation in CT images, 2023b.